# Review and Rebuttal: Zero-shot In-context Adversarial Learning for Improving Research Ideation

## Abstract

Recent studies highlight that the advancements in Large Language Models (LLMs) have opened up exciting possibilities for scientific discovery, where LLMs can assist researchers in generating novel hypotheses and ideas. In this work, we draw inspiration from Generative Adversarial Networks (GANs) and make the first effort to formalize the concept of zero-shot in-context adversarial learning and implement it through multi-LLM-agent interactions to improve the research ideation process. Our approach takes the best of two worlds: (1) by making in-context learning adversarial, the utilization of an LLM's vast parametric knowledge can be optimized; and (2) by keeping adversarial learning in context, we eliminate the need for bi-level optimization through additional model training. To evaluate the quality of the open-ended generation produced by LLMs, we develop a relative quality ranking metric, designed to serve as a proxy for human evaluation when human assessments are impractical or costly. Our findings demonstrate that zero-shot in-context adversarial learning significantly enhances idea generation across two dimensions. Specifically, with GPT-4o, the novelty of generated ideas improved by 21%, and feasibility of the ideas saw an impressive increase of 322%. These results underscore the transformative potential of zero-shot in-context adversarial learning in driving innovation and creativity within the research process.

## 1 Introduction

The rapid advancement of foundation models in machine learning has gained considerable momentum in recent years. Among these, large language models (LLMs) like GPT-4 (OpenAI, 2023) have introduced capabilities that set them apart from earlier machine learning models. A key milestone is their in-context learning ability, which allows LLMs to interpret and respond to user prompts without requiring additional task-specific training. This enables them to generalize across a wide variety of tasks, achieving state-of-the-art performance with minimal extra data (Brown et al., 2020). As a result, foundation models have redefined human-AI interactions, enabling accurate and fluent execution of tasks such as question answering, language translation, text and image generation, and even the creation of original content (Bubeck et al., 2023). These breakthroughs extend far beyond consumer applications, influencing critical domains like education (Moore et al., 2023) and healthcare (Yang et al., 2023a).

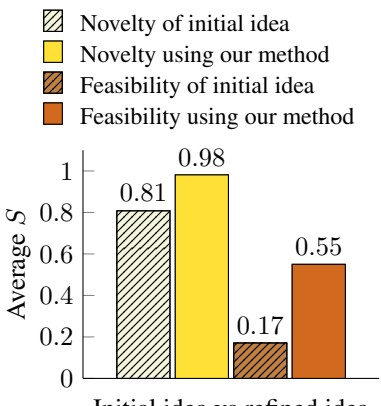

Figure 1: GPT-4o with zero-shot in-context adversarial learning improves the novelty and feasibility of the generated research ideas by a large margin.

Recently, breakthroughs in LLMs have sparked growing interest in academia, particularly regarding their potential to advance scientific research. Studies such as (Si et al., 2024) indicate that LLMs have the capability to generate research ideas comparable to human-level creativity. Numerous efforts have been made to explore various approaches for

utilizing LLMs in hypothesis generation, ranging from prompt engineering to supervised fine-tuning (Wang et al., 2023d; Baek et al., 2024; Yang et al., 2023b; Zhou et al., 2024; Boiko et al., 2023).

However, effectively harnessing the vast parametric knowledge within LLMs to improve research ideation remains a largely uncharted area. The challenge lies in the sheer complexity and scale of parametric knowledge, which is learned during the pre-training stage but may be underutilized in downstream tasks, especially when working with minimal user-provided context. To address this gap, we draw inspiration from Generative Adversarial Networks (GANs) (Goodfellow et al., 2020) and propose, for the first time, an adversarial learning framework in a zero-shot in-context learning setting. Unlike (Do et al., 2023), where prompts are optimized with adversarial elements to enhance in-context learning tasks, our formulation is more general, aiming to directly optimize performance on downstream tasks in an end-to-end manner, without requiring ground-truth data-label pairs in the context.

Our approach combines the advantages of both adversarial learning and in-context learning. On one hand, by leveraging adversarial learning, LLMs can more effectively utilize their parametric knowledge to respond to user query. On the other hand, keeping adversarial learning within an in-context learning framework simplifies the notoriously challenging convergence issues associated with bi-level optimization in adversarial training. We implement zero-shot in-context adversarial learning through a multi-LLM-agent interaction system. To scale the evaluation of generated ideas, we introduce a relative quality ranking metric, designed to approximate human-level evaluation in a customizable and fair manner. We experimented with the state-of-the-art LLMs using zero-shot in-context adversarial learning and measured the novelty and feasibility of the generated ideas using our metric (denoted $S$). The results in Figure 1 show that zero-shot in-context adversarial learning using GPT-4o significantly improves the novelty of the generated ideas by 21% and their feasibility by 322%.

In summary, our contributions are twofold:

- We formulate zero-shot in-context adversarial learning theory based on GANs, empowering LLMs to optimize the utilization of their parametric knowledge, thereby enhancing their ability to generate high-quality, suboptimal answers in response to user queries.

- We develop a novel relative quality ranking metric that provides a fair, flexible, and scalable approach to evaluating the quality of open-ended generation, serving as an effective proxy for human evaluation.

## 2 RELATED WORK

### 2.1 THEORETICAL FOUNDATIONS OF IN-CONTEXT LEARNING

Due to the black-box nature of LLMs, researchers have been drawing analogies to explain why and how in-context learning capabilities emerge in these models. Xie et al. (2021) argue that in-context learning can be viewed as a form of implicit Bayesian inference, arising when the pretraining corpus of LLMs contains documents with long-range coherence, forcing the model to infer latent concepts to generate coherent text. Olsson et al. (2022) provide a compelling argument that "induction heads" within transformer models play a crucial role in enabling in-context learning, as the model improves its predictions by processing more tokens within a sequence. Similarly, Dai et al. (2022) propose that the Transformer's attention mechanism implicitly performs meta-optimization, akin to gradient descent, where demonstrations create meta-gradients that fine-tune the model in context. Besides, TextGrad (Yuksekgonul et al., 2024) claims that the automatic differentiation can be performed via feedback for the generated answer provided by the LLMs.

### 2.2 LARGE LANGUAGE MODELS FOR SCIENTIFIC HYPOTHESIS GENERATION

Going beyond literature-based discovery (LBD), which primarily focuses on predicting pairwise relationships between discrete concepts (Wang et al., 2023c), recent research has started to explore the potential of foundation models, especially Large Language Models (LLMs) for scientific ideation (Si et al., 2024). For instance, the SciMON framework (Wang et al., 2023d) utilized historical scientific literature to fine-tune LLMs for generating hypotheses. In the social sciences, MOOSE (Yang et al., 2023b) employed multi-level LLM self-feedback to improve the discovery of scientific

hypotheses. Similarly, ResearchAgent (Baek et al., 2024) utilized LLMs to automatically generate and refine research problems, methodologies, and experimental designs, starting from a core paper and entity-centric knowledge graphs. Furthermore, (Zhou et al., 2024) proposed a prompting-based approach that iteratively generates hypotheses using LLMs guided by training examples. Beyond hypothesis generation, the Coscientist system described in (Boiko et al., 2023) equipped foundation models with tools such as internet search and code execution, demonstrating their potential for semi-autonomous experimental design and execution, particularly in chemical research.

## 2.3 MULTI-LLM-AGENT INTERACTIONS FOR IMPROVING TASK PERFORMANCE

While promising, a single LLM often struggles to generate novel insights after establishing an initial stance, even when that stance is incorrect, and lacks the necessary feedback mechanisms for rational refinement (Bubeck et al., 2023). Recent studies (Huang et al., 2023; Liang et al., 2023) have further highlighted that this challenge cannot be easily addressed through self-correction mechanisms(Madaan et al., 2024; Shinn et al., 2024). To overcome this limitation, recent works establish multi-LLM-agent frameworks through discussion (Chen et al., 2023; Lu et al., 2024), collaboration (Chih-Yao Chen et al., 2024; Zhang et al., 2023), and debate (Du et al., 2023; Liang et al., 2023; Subramaniam et al., 2024) to incorporate both internal and external knowledge, thereby enhancing model ability such as improving consistency (Xiong et al., 2023), evaluation (Chan et al., 2023; Wang et al., 2023a), and supervising other LLMs (Khan et al., 2024; Subramaniam et al., 2024). For example, Liang et al. (2023) introduced the Multi-Agent Debate framework, in which multiple LLMs engage in argument exchanges, with a judge overseeing the debate to determine the final solution. Similarly, Lu et al. (2024) proposed the LLM discussion framework which enhances the creativity through divergent thinking in the discussion phase and reach conclusion in the convergence phrase. Recent research has explored multi-LLM-agent interaction in various contexts, including scalable oversight (Kenton et al., 2024), translation (Liang et al., 2023), and knowledge reasoning (Wang et al., 2023b; Ma et al., 2024). Building upon these frameworks, we theoretically model the zero-shot in-context adversarial learning problem and extend it to the field of research ideation.

## 3 METHOD

In this section, we first present the theoretical framework for zero-shot in-context adversarial learning, aiming at optimizing the utilization of LLMs' parametric knowledge to perform user-specified tasks. In addition, we describe how this can be implemented through LLM-based agent interactions to enhance the generation of research ideas. Following this, we introduce a relative quality ranking-based metric designed to approximate human evaluation of the generated ideas.

### 3.1 ZERO-SHOT IN-CONTEXT ADVERSARIAL LEARNING FOR RESEARCH IDEA REFINEMENT

The goal of zero-shot in-context adversarial learning is to optimize the utilization of LLMs paramatric knowledge such that LLMs can generate a suboptimal answer with limited context provided by the user's query. To achieve this goal, we begin our formulation with Assumption 1:

**Assumption 1.** *We assume that given any user query $x$, there exists a static optimal answer $\dot{y}$, although the LLM may not explicitly generate $\dot{y}$ due to the discrete nature of its paramatric knowledge base $\{\theta\}$.*

The paramatric knowledge base $\{\theta\}$ of an LLM is obtained during the pre-training process of the given LLM and plays a crucial role in answering any user's query $x$. As the oracle answer $\dot{y}$ may not be directly achievable, the objective shifts to generating an approximation answer $\hat{y}$ which is sufficiently close to $\dot{y}$. Thus, we have Assumption 2:

**Assumption 2.** *Given a user query $x$, if an LLM generates a $\hat{y}$ from its paramatric knowledge base $\{\theta\}$, and $\hat{y}$ lies in the neighborhood $B$ of $\dot{y}$ with radius $\epsilon$, that is, $\hat{y} \in B_\epsilon(\dot{y})$, we posit that the LLM has optimized the use of its paramatric knowledge in response to the user's query $x$, yeilding suboptimal answer $\hat{y}$.*

To optimize the generation of $\hat{y}$, we foumulate the objective inspired by Generative Adversarial Networks (GANs) (Goodfellow et al., 2020). Similar to GANs, the objective of zero-shot in-context

adversarial learning is framed as a minimax game between two models: a Generator $G$ and a Discriminator $D$. The Generator's goal is to generate an answer $\hat{y}$ to approach $B_\epsilon(\dot{y})$, while the Discriminator is tasked with determining whether $\hat{y}$ belongs to $B_\epsilon(\dot{y})$. Therefore, the objective function of this minimax game can be defined as follows:

$$V(G, D) = \min_G \max_D \mathbb{E}_{\dot{y} \in B_\epsilon(\dot{y})}[\log D(\dot{y})] + \mathbb{E}_{x \sim p_x(x)}[\log(1 - D(G(x)))]$$

$$s.t., \begin{cases} \theta_D^* = \arg\max_{\theta_D} \mathbb{E}_{\dot{y} \in B_\epsilon(\dot{y})}[\log D(\dot{y})] + \mathbb{E}_{x \sim p_x(x)}[\log(1 - D(G(x)))] \\ \theta_G^* = \arg\min_{\theta_D} \mathbb{E}_{\dot{y} \in B_\epsilon(\dot{y})}[\log(1 - D(G(x)))] \end{cases} \quad (1)$$

where:

- $\mathbb{E}_{\dot{y} \in B_\epsilon(\dot{y})}[\log D(\dot{y})]$ represents the expected log-probability that the Discriminator assigns to the optimal answer $\dot{y}$, with the goal of maximizing this term so that the Discriminator can correctly reject any approximation $\hat{y}$ in the $B_\epsilon(\dot{y})$.
- $\mathbb{E}_{x \sim p_x(x)}[\log(1 - D(G(x)))]$ represents the expected log-probability that the Discriminator assigns to generated answer $G(x)$, where $G(x) = \hat{y}$, and $x$ is a user query sampled from the user query distribution $p_x(x)$. The Generator aims to minimize this term, trying to convince the Discriminator to accept $\hat{y} \in B_\epsilon(\dot{y})$.

During this adversarial process, the Generator aims to minimize $\log(1 - D(G(x)))$, meaning it tries to convince the Discriminator to accept $\hat{y} \in B_\epsilon(\dot{y})$. Conversely, the Discriminator aims to maximize both $\log D(\dot{y})$ for the optimal answer $\dot{y}$ and $\log(1 - D(G(x)))$ for the generated answer $\hat{y}$. According to Proposition 2 in Goodfellow et al. (2020), if $G$ and $D$ have enough capacity, during the optimization process, $\hat{y}$ converges to $\dot{y}$. According to Theorem 1 in Goodfellow et al. (2020), the global minimum of the objective function is reached if and only if $\hat{y} = \dot{y}$. Though in open-ended generation tasks for LLMs it's challenging to generate $\hat{y} = \dot{y}$, achieving $\hat{y} \in B_\epsilon(\dot{y})$ remains plausible and practical. Note that $\epsilon$ is likely to vary from model to model.

The objective function defined in Formula 1 can be optimized through in-context learning in LLMs, so no actual model parameters are updated throughout the process. Instead, this optimization is achieved by forcing $D$ and $G$ to search in their parametric knowledge base $\{\theta_D\}$ and $\{\theta_G\}$ to get $\theta_D^*$ and $\theta_G^*$, respectively.

To implement zero-shot in-context adversarial learning for research idea generation and refinement, we employ a multi-agent interaction system using LLMs. There are three agents in the system, each agent plays a unique role in the objective function and will be introduced in the following subsections. The overview of this system for research idea refinement is shown in Figure 2. In general, once the user provides a context $x$, the proposer agent acts as the generator $G$ to generate and refine idea $\hat{y}$, the reviewer agent serves as the optimizer by providing the gradient $r$, and the area chair agent functions as the discriminator $D$ in the objective function, continuing this process until the minimax game converges to equilibrium. For simplicity, the minimax game is shown to converge at the 4th iteration in Figure 2; however, in practice, the steps illustrated in iterations 2 and 3 may repeat multiple times, and additional iterations may be required for the minimax game to fully converge.

### 3.1.1 RESEARCH IDEA PROPOSER

The research idea proposer agent acts as the Generator $G$ in the objective function. Its role is to generate and iteratively refine the research idea $\hat{y}$, striving to approach the optimal idea $\dot{y}$. At the beginning of the minimax game, the proposer agent is profiled as a domain expert researcher and generates an initial idea $\hat{y}_0$ based on the user query $x$. In subsequent iterations of the minimax game, the proposer agent is tasked with refining the idea based on feedback from the reviewer agent. Therefore, at the $i$-th iteration, the proposer agent updates $\hat{y}$ via:

$$\begin{cases} \theta_{i,G} = \theta_{i-1,G} - \eta r_i \\ \hat{y}_i = G(\hat{y}_{i-1}; \theta_{i,G}) \end{cases} \quad (2)$$

where $r_i = \nabla_\theta V(G, D; \theta_i)$ is the "textual gradient" for updating the parametric knowledge for refining $\hat{y}$, which is provided by the research idea reviewer agent through its feedback and will be

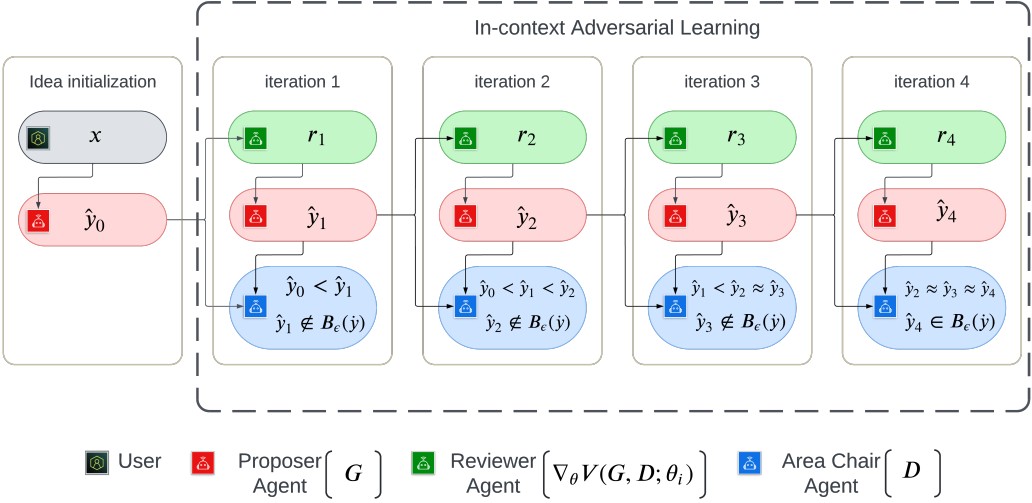

Figure 2: The overview of in-context adversarial learning via LLM-based agent interactions for research idea generation and refinement.

introduced in 3.1.2 with more details. The learning rate $\eta$ is dynamically and implicitly determined by the generator $G$. We demonstrate all the prompt templates for research idea proposer agent in Fig. 5, Fig. 8, and Fig. 9 in the Appendix.

### 3.1.2 RESEARCH IDEA REVIEWER

The research idea reviewer agent offers feedback $r$ on the proposer's idea $\hat{y}$ as the "textual gradient" $\nabla_\theta V(G, D; \theta_i)$ that guides the proposer agent in refining the idea. Compared to traditional numerical gradients, "textual gradient" takes the form of text, making them more interpretable while still functioning similarly to numerical gradients in optimizing downstream tasks (Yuksekgonul et al., 2024). At the beginning of the minimax game, the reviewer agent is also profiled as a domain expert researcher, but its primary task is to review the ideas and offer feedback, rather than generate ideas. At each iteration step $i$, the reviewer agent is asked to critique the current idea based on quality indicators such as novelty or feasibility, as specified by the user. It provides constructive feedback that the proposer agent can leverage to refine the idea. The prompt templates for the research idea reviewer agent are presented in Fig. 6 and Fig. 10 in the Appendix.

### 3.1.3 AREA CHAIR

The area chair agent functions as the primary Discriminator $D$ in the minimax game. Similar to the other agents, it is initially profiled as a domain expert researcher. At each iteration step $i$, the area chair agent is tasked with identifying the improvements between the current idea $\hat{y}_i$ and the previous idea $\hat{y}_{i-1}$. Although $\hat{y}_i$ is represented as text rather than numbers, we use the symbol "$<$" to indicate cases where the area chair agent detects significant improvements in the new idea compared to the previous one. Conversely, "$\approx$" denotes situations where the area chair agent does not identify any substantial improvements. As is shown in Figure 2, if significant improvements between the two ideas can be identified by the area chair agent, that is, $\hat{y}_{i-1} < \hat{y}_i$, it suggests that $\hat{y}_i$ might not belong to $B_\epsilon(\dot{y})$. Thus, further refinement on $\hat{y}_i$ with respect to user-specified quality indicators is necessary. However, if the area chair agent consistently determines that there is no substantial

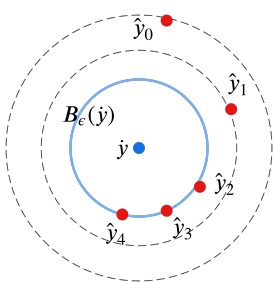

Figure 3: Idea evolution dynamics.

improvement between the latest and previous iterations, that is, $\hat{y}_{i-2} \approx \hat{y}_{i-1} \approx \hat{y}_i$, we posit that the optimization has converged to equilibrium, implying that both $\hat{y}_{i-1}$ and $\hat{y}_i \in B_\epsilon(\dot{y})$, with $\epsilon$ being the implicit distance between $\hat{y}_i$ and $\dot{y}$. In this case, $\hat{y}_i$ is considered as the final suboptimal research idea. The prompt templates for the area chair agent to fulfill its role as the Discriminator $D$ are presented in Fig. 7 and Fig. 11 in the Appendix.

Figure 3 illustrates the evolution dynamics of the generated research idea $\hat{y}$. As the area chair agent continues to identify that $\hat{y}_1$ and $\hat{y}_2$ carry significant improvements over their respective predecessors, it becomes necessary for the proposer agent to further refine these ideas to convince the area chair agent that the subsequent idea $\hat{y}_3$ belongs to $B_\epsilon(\hat{y})$. From $\hat{y}_2$ to $\hat{y}_4$, the area chair agent is convinced that no further improvements can be identified, leading the minimax game to converge to equilibrium. Consequently, the final idea $\hat{y}_4 \in B_\epsilon(\dot{y})$ is selected as the final suboptimal research idea. For ease of illustration, $\hat{y}$ in Figure 3 converges at the 4th iteration, but in practice, it may take more iterations for $\hat{y}$ to converge.

### 3.2 RELATIVE QUALITY RANKING FOR APPROXIMATING HUMAN-LEVEL HYPOTHESIS EVALUATION

Although human judgment remains the gold standard for evaluating open-ended text generation, the Natural Language Processing community has been actively developing scalable alternatives to approximate human evaluation, as the labor involved in human evaluation is often costly and impractical in many cases. Recent studies have explored the use of LLMs as autoraters (Chiang & Lee, 2023; Liu et al., 2023; Bubeck et al., 2023; Fu et al., 2024; Vu et al., 2024; Gu & Krenn, 2024). These studies show that the correlation between human evaluators and LLM autoraters positions LLMs as a promising alternative for large-scale assessments for open-ended generation. To automate the evaluation of the generated ideas, we develop a relative ranking-based metric designed to assess idea quality in a fair and customizable manner. This metric can be customized to accommodate various quality indicators, such as novelty, feasibility, or any other criteria specified by the user, as long as a target research idea and the context used to generate this target idea are available. The target idea can either be generated by the user or selected from existing literature. Compared to the winrate, our metric offers a more granular measurement (Zheng et al., 2023). Please refer to Section A.3.1 in the Appendix for more discussions.

For a given context used to generate a set of research ideas, we use GPT-4o to rank the quality of all the ideas (both generated ideas and the target idea) based on user-specified quality indicators, without revealing which idea is the target research idea. GPT-4o is prompted to assess the ideas based on its understanding of quality indicators such as novelty and feasibility, and then rank them accordingly. The prompt template used to achieve this is shown in Figure 12 in the Appendix. The position of the target research idea within the ranked list of ideas reflects the quality of the generated ideas with respect to the specified quality indicators. Intuitively, if the target idea ranks higher on the list, this suggests that the generated ideas are of lower quality compared to the target idea. Conversely, if the generated ideas rank higher than the target idea, it indicates that the generated ideas may be of better quality. Given a target idea $t$ and $n$ generated ideas based on the given context, let $n_t$ denote the rank of $t$ among the target idea and the generated ideas. The relative quality ranking $S$ of the generated ideas is computed as follows:

$$S = \frac{n_t - 1}{n} \tag{3}$$

Intuitively, $S \in [0, 1]$. If the target idea ranks first on the list, then $n_t = 1$, yielding $S = 0$, which indicates that all generated ideas are worse than the target idea. Conversely, if the target idea ranks below all the generated ideas, that is, $n_t = n + 1$, then $S = 1$, indicating that all generated ideas are superior to the target idea. To ensure fair comparison across different idea generation strategies, it is important to generate the same number of research ideas $n$ for all compared strategies.

## 4 EXPERIMENTS

The primary objective of our experiments is threefold: (1) to assess whether zero-shot in-context adversarial learning enhances the quality of research ideas generated by LLMs, (2) to examine how research ideas evolve and converge during the ideation process, and (3) to evaluate the contribution of each component in our multi-agent system to the overall performance.

To achieve this, we construct a dataset of high-quality biomedical papers and their references. Research ideas from these papers serve as target ideas, which we compare to the LLM-generated ideas using the relative quality ranking metric introduced in Section 3.2. The references provide contextual information which simulate user queries that begin generation of research ideas in our system.

We use zero-shot in-context adversarial learning to generate research ideas with enhanced novelty and feasibility. Our results highlight our method's effectiveness and clarify the contribution of each agent in our method's ideation process.

## 4.1 DATASET

To evaluate the effectiveness of zero-shot in-context adversarial learning in enhancing research idea generation we constructed a dataset designed for this specific task. We gathered a set of "target papers"; high-quality biomedical research papers published in 2024. We denote a target paper's research idea as $t_i$, where $i$ indexes the target paper out of the total $m$ papers in our dataset. The ideas of the target papers represent the "gold standard" for comparing the generated research ideas through the relative quality ranking score.

In order to simulate the initial human query to the system we also collect the background information that informed each paper's research. This background information consists of the abstracts from the reference papers cited by the target paper, which we represent as $x_i = \{b_1, \ldots, b_{k_i}\}$, where $k_i$ is the total number of references for the $i$-th paper.

By linking the target papers to their reference papers, we build a comprehensive dataset that provides LLMs with the foundational context necessary to generate research ideas that stem from the same background information from which the target papers were inspired. Formally, we represent the dataset as $\{x_i, t_i\}_{i=1}^m$ , where each data point consists of a target paper's research idea $t_i$ and its corresponding background information $x_i$ for $m$ papers.

We sourced target papers from top biomedical venues ranked by Google Scholar, using the Semantic Scholar API (Kinney et al., 2023). To ensure a high standard, we included papers from top-tier venues with at least one citation or those from other recognized venues that have garnered at least 20 citations. Duplicate entries were removed, and we only included papers that contained all essential data fields, such as abstracts, to ensure dataset completeness. For our experiments, we gathered $m = 500$ target papers and their corresponding background information.

## 4.2 EXPERIMENTAL SETUP

To evaluate the effectiveness of zero-shot in-context adversarial learning in improving research ideation, we set up the proposer, reviewer and area chair agents to engage in structured interactions through an agent discussion framework. This setup allows us to test whether the framework enhances the novelty and feasibility of the generated research ideas.

**Agent Initialization.** We initialize the agents with meta prompts (Liang et al. (2023)) to make sure each agent understands its role and which field it specializes in. The meta prompts specify the agent's role—whether it is the proposer, reviewer, or area chair—detailing what the role entails, which {research_area} the agent specializes in, and which {quality_indicator} (novelty or feasibility) the agent is tasked with improving or judging. All the prompt templates along with the algorithm for agent interactions are provided in the Appendix. This initialization underpins the multi-agent system's ability to enhance research ideation using zero-shot in-context adversarial learning.

**Idea Initialization.** To simulate how humans would query this system with some background context to generate the initial research idea, we set the initial query to be a set of background information $x_i = \{b_1, \ldots, b_{k_i}\}$ for a target idea $t_i$. This background information is given to the proposer agent to generate the initial research idea $\hat{y}_0$. The initial idea is then sent through our system to improve the idea's novelty or feasibility with zero-shot in-context adversarial learning.

**Baselines.** To evaluate the effectiveness of our zero-shot in-context adversarial learning system, we compare its performance against two baselines: (1) the initial idea baseline and (2) the self-reflection baseline. We measure improvement based on two key quality indicators—novelty and feasibility—using the relative ranking quality score $S$ described in Section 3.2.

- **Initial idea**: This baseline is the initial idea $\hat{y}_0$ generated by the proposer agent when given a set of background information $x_i = \{b_1, \ldots, b_{k_i}\}$ for a target idea $t_i$. GPT-4o is the backbone LLM for this baseline.
- **Self-reflection**: For this baseline, the self-reflection method is used. The same initial ideas $\hat{y}_0$ are iteratively improved through self-evaluation (Madaan et al., 2024; Liang et al., 2023),

where the proposer agent reflects on its own generated research ideas and modifies them without external interaction. The agent stops iterating once it thinks its ideas stopped improving. GPT-4o is also the backbone LLM for this method.

In the main experiment, we compare our method with baselines and evaluate performance using GPT-4o, GPT-4o Mini, and GPT-3.5 Turbo as backbone models. For both the self-reflection baseline and our method, we generate results in two cases: one focusing on improving novelty and the other on feasibility. In each case, three research ideas are generated for each target idea. These ideas, along with the target paper's idea, are ranked based on novelty and feasibility using the relative quality ranking score $S$ (Equation 3). Although the same ranking mechanism is used, the rankings are computed separately for novelty and feasibility, allowing us to evaluate each dimension independently. A prompt template is used to rank research ideas for novelty or feasibility without knowing whether the ideas are human or LLM generated, ensuring fairness (see Figure 12 and the Appendix for more details). All rankings are computed using GPT-4o, the highest-capacity model, regardless of the backbone model used for generation. Finally, $S$ is averaged across all target papers in the dataset.

### 4.3 MAIN RESULTS

| Method | Base Model | Average $S$ (Novelty) | Average $S$ (Feasibility) |
|---|---|---|---|
| Initial idea | GPT-4o | 0.808 | 0.171 |
| Self-reflection | GPT-4o | 0.952 | 0.342 |
| Our method | GPT-3.5 Turbo | 0.963 | 0.589 |
| | GPT-4o Mini | 0.960 | **0.762** |
| | GPT-4o | **0.981** | 0.550 |

Table 1: Main experiment results. For each method, we report the average of the relative quality ranking scores, denoted as *Average S*, for novelty and feasibility.

Our main experiment demonstrates that zero-shot in-context adversarial learning significantly enhances the quality of research ideas generated by LLMs. Specifically, our results reveal that research ideas generated through this method consistently rank higher in both novelty and feasibility compared to high-quality human-generated ideas since the average $S$ for both novelty and feasibility are above $0.5$. Furthermore, our method allows for the lower-capacity LLMs to outperform their higher-capacity counterparts, a surprising result that highlights the strength of zero-shot in-context adversarial learning. Table 1 summarizes these improvements, showing our method's substantial gains in both novelty and feasibility of research ideas.

Our method demonstrates significant improvements over both baselines. with GPT-4o, we observe a 21% increase in novelty and an impressive 322% improvement in feasibility compared to the initial ideas. Specifically, when zero-shot in-context adversarial learning is used to generate novel research idea, our method with GPT-4o gets an impressive average relative quality ranking score of 0.981. Similarly, to generate feasible research ideas, our method gets a score of 0.762 when using GPT-4o Mini. These are not only higher than what the baselines yield, but it also signifies that a majority of research ideas produced with our method outrank human generated research ideas in novelty and feasibility.

Remarkably, zero-shot in-context adversarial learning also makes it possible for lower-capacity LLMs to outperform higher-capacity ones. Using GPT-4o Mini and GPT-3.5 Turbo, our method exceeds baseline methods with GPT-4o. Our method enables lower-capacity models to optimize using their parametric knowledge, allowing them to rival or exceed the performance of higher-capacity LLMs in specific tasks like research idea generation.

These results validate the effectiveness of zero-shot adversarial in-context learning. We observe that refining research ideas with out method outperforms self-reflection, a strong method for refining LLM outputs. Moreover, the success of smaller models like GPT-4o Mini and GPT-3.5 Turbo in outperforming GPT-4o underscores the potential of zero-shot in-context adversarial learning, even with lower-capacity models.

## 4.4 CONVERGENCE ANALYSIS

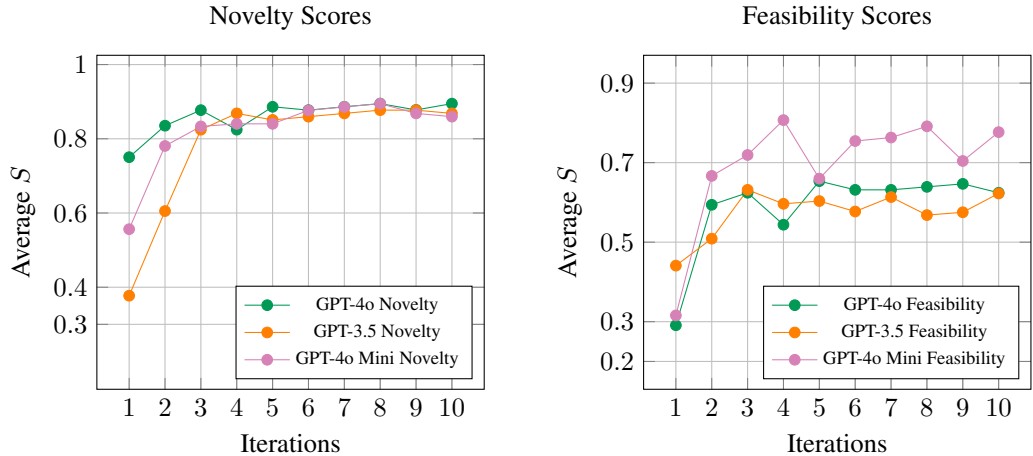

Figure 4: Evolution of research ideas' novelty (left) and feasibility (right) as number of iterations increases.

| Backbone LLM | Improvement Type | Median Iterations | Cvg. < 10 |
|---|---|---|---|
| GPT-4o | Novelty | 6 | 0.980 |
| | Feasibility | 5 | 0.963 |
| GPT-3.5 Turbo | Novelty | 10 | 0.254 |
| | Feasibility | 10 | 0.348 |
| GPT-4o Mini | Novelty | 10 | 0.005 |
| | Feasibility | 10 | 0.084 |

Table 2: Convergence statistics of our method from the main experimental results when improving novelty and feasibility. We report the median number of iterations until convergence, as well as the proportion of runs that reach convergence within 10 iterations (Cvg. < 10).

In this section, we analyze how the quality of research ideas evolve during the iterative process of zero-shot in-context adversarial learning. Figure 4 illustrates improvements in both novelty and feasibility as iterations increase. We show that research idea quality consistently increases in both dimensions before converging after a few iterations, demonstrating the effectiveness of our method.

The initial improvement in quality shows that the reviewer agent's feedback $r_i$ generates a gradient $\nabla_\theta V(G, D; \theta_i)$, which helps the proposer agent to search for its paramatric knowledge $\theta_{i,G}$ such that it can refine the research ideas, moving them closer to an optimal neighborhood $B_\epsilon(\hat{y})$. After a few iterations, the quality of the research idea converges, signaling that the generated ideas are within an optimal neighborhood.

How quickly the area chair detects convergence depends on the backbone LLM. With GPT-4o, convergence is achieved within 10 iterations over 96% of the time, as shown in Table 2, with a median of 6 iterations for novelty and 5 for feasibility. This aligns with Figure 4, where improvements in idea quality converge after a few iterations.

In contrast, lower-capacity LLMs like GPT-4o Mini and GPT-3.5 Turbo detect convergence much later, with a median of 10 iterations. GPT-3.5 Turbo detects convergence within 10 iterations less than 40% of the time, and GPT-4o Mini less than 10%. The lower capacity of these LLMs makes it difficult for the area chair agent to see when ideas stop improving, delaying the detection of convergence.

Our findings show that zero-shot in-context adversarial learning iteratively improves research ideas' novelty and feasibility, with convergence detection depending on the backbone LLM. Higher-capacity models like GPT-4o detects convergence earlier, while lower-capacity models like GPT-3.5 Turbo and GPT-4o Mini may suffer from early detiection of the minimax game's convergence.

GPT-4o's rapid convergence indicates efficient halting by the area chair agent when ideas reach the optimal neighborhood $B_\epsilon(\dot{y})$.

## 4.5 ABLATION STUDY

In this section, we evaluate the impact of removing key components of our zero-shot in-context adversarial learning system by performing ablation experiments. Our analysis focuses on how the absence of the area chair and reviewer agents affects the system's ability to refine research ideas and converge to the optimal idea neighborhood $B_\epsilon(\dot{y})$. GPT-4o is used as the agents' backbone LLM in the ablation study, conducted on a smaller dataset with $m = 100$ target papers. The results are summarized in Table 3.

| Ablation | Average $S$ (Novelty) | Median Iters. | Average $S$ (Feasibility) | Median Iters. |
|---|---|---|---|---|
| w/o area chair | 0.974 | 7 | 0.220 | 7 |
| w/o reviewer | 0.967 | 9 | 0.505 | 6 |
| Our method | **0.983** | **6** | **0.521** | **5** |

Table 3: Ablation study results showing the impact of our system without (w/o) the area chair or reviewer agent on the novelty and feasibility of generated research ideas. We report the average relative quality ranking scores (*Average $S$*) and median iterations (iters.) until convergence.

In the absence of the area chair agent, the system lacks the discriminator tasked with determining whether the generated idea $\hat{y}$ lies within the neighborhood $B_\epsilon(\dot{y})$ of the optimal idea. Without this key component, we fix the number of iterations to seven, since the median iterations needed for our method to converge is below seven (Table 2). The results show a drop in the novelty and a major decline in the feasibility for the research ideas, indicating that the area chair plays a crucial role in ensuring whether the generated ideas fall within $B_\epsilon(\dot{y})$. Without the area chair to stop the process when an idea is within the optimal neighborhood, many ideas fail to reach the optimal idea neighborhood, especially in terms of feasibility. This supports our theoretical framework that convergence depends on the area chair's ability to assess when further refinement is unnecessary.

The reviewer agent provides the essential gradient $r_i = \nabla_\theta V(G, D; \theta_i)$, enabling the generator to refine the idea iteratively. Removing the reviewer increases the number of iterations required to reach convergence since the proposer agent lacks effective feedback to guide the search for $\theta_G^*$. Without the reviewer, both novelty and feasibility suffer. This aligns with the theoretical formulation that reviewer feedback is essential for approximating the gradient necessary to update the generator's parameters and consequently moving $\hat{y}$ toward the optimal neighborhood $B_\epsilon(\dot{y})$.

Our full method, which includes all three agents, demonstrates superior performance in both novelty and feasibility. The average relative quality ranking scores are the highest, and the system converges faster than without the area chair or reviewer agent. These results reinforce the importance of each agent in the minimax game.

The ablation study highlights the critical roles of the area chair and reviewer agents in the zero-shot in-context adversarial learning framework. Without these agents, the system either converges more slowly or fails to consistently produce ideas that approach $B_\epsilon(\dot{y})$. Please refer to Section A.3 the Appendix for more experiments and discussions.

## 5 CONCLUSION

In this work, we formulated zero-shot in-context adversarial learning and implemented it through a multi-LLM-agent interaction system to enhance the scientific research ideation process. Additionally, we developed a relative quality ranking metric to evaluate the generated ideas in a customizable and fair manner, serving as a proxy for human evaluation. Our promising results demonstrate that in-context adversarial learning not only improves scientific ideation but also holds potential for enhancing other tasks involving user interaction with LLMs. We hope this work paves the way for greater adoption of LLMs in scientific discovery and advances in in-context learning theory for improving LLM performance.

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

# A APPENDIX

## A.1 MULTI-LLM-AGENT SYSTEM IMPLEMENTATION DETAILS

This section provides further details on the implementation of zero-shot in-context adversarial learning via multi-LLM-agent interaction.

The proposer, reviewer, and area chair agents interact iteratively following Algorithm 1.

Each agent has a meta prompt defining its role and task. Figures 5, 6, and 7 illustrate the meta prompts for the proposer, reviewer, and area chair, respectively. These prompts take in user-defined {research_area} and {quality_indicator} hyperparameters, which specify the research field and the aspect of research ideas to improve. Additionally, the proposer and reviewer agents' prompts take a {quality_indicator_traits} hyperparameter, listing the traits relevant to the {quality_indicator}.

Each agent also uses task-specific prompt templates. The proposer generates an initial idea $\hat{y}_0$ using the template in Figure 8, inputting the {research_area} and a given target paper's reference paper abstracts $\{b_1, \ldots, b_{k_i}\}$ with the prompt template parameters {background_paper_1_abstract}, ..., {background_paper_k_abstract}. After receiving feedback from the reviewer, the proposer revises the idea using the template in Figure 9 to improve the {quality_indicator}. The reviewer provides feedback with the template in Figure 10, considering both the {quality_indicator} and {quality_indicator_traits}. The area chair then assesses whether the proposer successfully improved its idea's {quality_indicator} using the template in Figure 11.

In our experiments, we set {research_area} to biomedical and evaluate the system on two {quality_indicator} values: novelty and feasibility. For novelty, we set {quality_indicator_traits} to be "creativity of the hypothesis, innovation of the approach, disruptiveness, originality, conceptual shift, and addressing a research gap." For feasibility we set {quality_indicator_traits} to be "accessibility of resources, simplicity of method, data availability, time and cost efficiency, scalability, and practicality." Additionally, in Algorithm 1, we set the maximum number of iterations, max_iters to 10 for all experiments.

---

**Algorithm 1** Algorithm for zero-shot in-context adversarial learning for research idea generation via multi-LLM-agent interactions.

---

**Input:** User defined {quality_indicator}, {quality_indicator_traits}, {research_area}, background context $\{b_1, \ldots, b_{k_i}\}$, maximum iterations max_iters
**Output:** Final research idea $p_{i+1}$
1: **Step 1: Initialization**
2: Initialize proposer, reviewer, and area chair agents based on {quality_indicator}, {quality_indicator_traits}, and {research_area}
3: $i \leftarrow 0$
4: Generate initial research idea $p_i \leftarrow \text{proposer}(\{b_1, \ldots, b_{k_i}\})$
5: $r_i \leftarrow \text{reviewer}(p_i)$
6: **Step 2: Iterative Improvement Process**
7: **while** $i < \text{max\_iters}$ **do**
8:     Generate new research idea $p_{i+1} \leftarrow \text{proposer}(r_i)$
9:     Review new idea $r_{i+1} \leftarrow \text{reviewer}(p_{i+1})$
10:     $stop \leftarrow \text{area\_chair}(p_i, p_{i+1})$
11:     **if** $stop$ is True **then**
12:         **break**
13:     **end if**
14:     $i \leftarrow i + 1$
15: **end while**
16: **Step 3: Return Final Idea**
17: **Return** final research idea $p_{i+1}$

---

**Proposer Meta Prompt Template.**

You are a {research_area} researcher proposing research ideas. Your role is to create a research idea and refine the idea if you receive feedback. A reviewer will review your research idea based on its {quality_indicator} and give you feedback. You should try your best to improve the idea based on the reviewer's feedback and your expertise, especially paying attention to the idea's {quality_indicator_traits}.

Figure 5: Meta prompt template for the proposer agent to inform the agent of its role and responsibility.

**Reviewer Meta Prompt Template.**

You are an experienced {research_area} researcher reviewing research ideas. Your role is to receive a research idea and try your best to give constructive criticism about the idea's {quality_indicator} so that the proposer can review your feedback and improve the idea's {quality_indicator} as much as possible. When reviewing, focus on the idea's {quality_indicator_traits}.

Figure 6: Meta prompt template for the reviewer agent to inform the agent of its role and responsibility.

**Area Chair Meta Prompt Template.**

You are an area chair for a high-impact {research_area} conference. You will receive a proposer's prior research idea and the proposer's revised research idea based on a reviewer's feedback. Your role is to try your best to identify any improvement in the revised idea and determine whether the revised idea has a significant improvement in {quality_indicator}.

Figure 7: Meta prompt template for the area chair agent to inform the agent of its role and responsibility.

**Proposer Agent Prompt Template for Generating Initial Research Ideas**

You are a {research_area} researcher. You are tasked with creating a hypothesis or research idea given some background knowledge. The background knowledge is provided by abstracts from other papers.

Here are the abstracts:

Abstract 1:{background_paper_1_abstract}
Abstract 2:{background_paper_2_abstract}
......
Abstract k:{background_paper_k_abstract}

Using these abstracts, reason over them and come up with a novel hypothesis. Please avoid copying ideas directly, rather use the insights to inspire a novel hypothesis in the form of a brief and concise paragraph.

Figure 8: Prompt template for the proposer agent to generate an initial research idea based on research paper abstracts as background context.

**Proposer Agent Prompt Template.**

```
{reviewer_agent_feedback}
```

Based on the reviewer's feedback regarding the previous research idea's `{quality_indicator}`, generate a revised and improved research idea using the following format:

Title: [A brief, focused title]

Problem: [The core issue or gap being addressed]

Objective: [The main goal or research question]

Hypothesis: [The hypothesis being tested or explored]

Method: [The approach or methodology]

Expected Impact/Findings: [The anticipated outcomes or contributions].

Please only respond with the improved research idea returned in the format provided above. Do not respond with anything irrelevant.

Figure 9: Prompt template for the proposer agent to generate a revised research idea based on the reviewer agent's feedback.

**Reviewer Agent Prompt Template.**

You will receive the proposer's research idea. Try your best to give the best constructive criticism on the research idea's `{quality_indicator}` so that the proposer can improve the idea's `{quality_indicator}` as much as possible. In your response, please explain why the research idea lacks in `{quality_indicator}`, specifically considering the idea's `{quality_indicator_traits}`. Here is the proposer's research idea: `{research_idea}`.

Figure 10: Prompt template for the reviewer agent to give constructive criticism and feedback to the proposer agent for its generated research idea.

**Area Chair Agent Prompt Template.**

Here is the proposer's prior idea: `{prior_research_idea}`

The proposer's revised idea: `{new_research_idea}`

You will compare the proposer's prior and revised ideas in this round and try your best to determine what improvement has been made in the revised idea and answer whether the revised idea has significantly improved in `{quality_indicator}`.

Please answer in a Python Dictionary with the following format:
`{"Is there a significant improvement?": "Yes" or "No"}`

Please strictly output in the Python Dictionary format; do not output irrelevant content.

Figure 11: Prompt template for the area chair agent to determine whether proposer's research ideas has improved.

## A.2 RELATIVE QUALITY RANKING IMPLEMENTATION DETAILS

In Section 3.2, we introduce the relative quality ranking metric, which evaluates LLM-generated research ideas based on a specified {quality_indicator}. The term $n_t$ in Equation 3 represents the rank of a human-generated target idea $t$ when compared to $n$ other LLM-generated ideas.

To compute $n_t$, we use GPT-4o with the prompt template shown in Figure 12. This template takes the {quality_indicator}, the {target_paper_idea}, and the $n$ LLM-generated ideas ({generated_idea_1},..., {generated_idea_n}) as inputs.

---

**Prompt template used to rank research ideas based on user specified quality indicators**

You are a reviewer tasked with ranking the quality of a set of research ideas based on their {quality_indicator}. The idea with the highest {quality_indicator} should be ranked first.

Please rank the following hypotheses in the format: 1. Hypothesis (insert number):(insert brief rationale)
2. Hypothesis (insert number):(insert brief rationale)
3. Hypothesis (insert number):(insert brief rationale)
......
n. Hypothesis (insert number):(insert brief rationale)

Please rank the following hypotheses:
Hypothesis 1: {target_paper_idea}
Hypothesis 2: {generated_idea_1}
Hypothesis 3: {generated_idea_2}
......
Hypothesis n: {generated_idea_n}

---

Figure 12: Prompt template used to rank research ideas based on user specified quality indicators.

To ensure a fair comparison between the LLM-generated ideas and the target paper's idea, we extract the core research idea from the target paper's abstract using GPT-4o with a customized prompt. Abstracts often include extraneous details, such as results or technical specifics, which may not reflect the central idea. To avoid bias in the ranking, we use a prompt that summarizes the main research idea, aligning with the style in which the LLM generates ideas. The prompt template for this extraction is shown in Figure 13. This process ensures an equitable ranking of the target paper's idea alongside the LLM-generated ideas.

---

**Paper Abstract Summary Prompt Template.**

Write a concise summary of the following paper abstract, proposing a research idea based on the abstract's content. Format the summary using the following structure, and if a field does not exist in the abstract, write "NONE" for that field:

Title: [A brief, focused title]

Problem: [The core issue or gap being addressed]

Objective: [The main goal or research question]

Hypothesis: [The hypothesis being tested or explored]

Method: [The approach or methodology]

Expected impact / findings: [The anticipated outcomes or contributions]

Abstract: `{target_paper_abstract}`
Summary:

---

Figure 13: Prompt template for summarizing a target paper's abstract into a research idea.

## A.3 ADDITIONAL EXPERIMENTS AND DISCUSSION

In this section, we offer further experiments and noteworthy discussions. We evaluate and discuss the validity of our automatic evaluation of research ideas using the proposed relative quality ranking metric with GPT-4o. Additionally, we present extra experiments using LLMs as the base models for our method and discuss the cost of generating research ideas using our method.

### A.3.1 RELATIVE QUALITY RANKING

*Alignment with Human Judgment*

In SCIMUSE, the authors collaborated with over 100 research group leaders across diverse domains to rank more than 4,400 research ideas generated by their SCIMUSE system(Gu & Krenn, 2024). Their findings revealed that LLM-based ranking, specifically using GPT-4o, aligns closely with human expert evaluations, achieving a top-1 precision of 51% and a top-5 precision of 46.7%. These results highlight the feasibility of using LLM-driven ranking as a scalable proxy for human evaluation, particularly when assessing large volumes of research ideas across various fields.

To evaluate the alignment between GPT-4o and humans in assessing research ideas within our context, we conducted a human study. We selected 10 sets of research ideas focused on novelty and 10 sets focused on feasibility, generated using our proposed adversarial in-context learning. Each set included three generated ideas and their respective target paper idea.

| | Average $D(S)\downarrow$ |
|---|---|
| **Novelty** | 0.1 |
| **Feasibility** | 0.3 |

Table 4: GPT-4o's alignment with human researchers in ranking target paper research ideas relative to generated research ideas.

We recruited 10 researchers to rank the ideas in each set based on either novelty or feasibility, depending on the focus. The researchers were unaware of which ideas were generated and which originated from the target paper. We then compared the difference between relative quality ranking given by human researchers and GPT-4o $D(S)$:

$$D(S) = |S_{\text{Human}} - S_{\text{GPT-4o}}| \tag{4}$$

where $S_{\text{Human}}$ is the relative quality ranking from human researchers calculated using Formula (3) and $S_{\text{GPT-4o}}$, similarly, is the relative quality ranking from GPT-4o. The Table 4 shows the average $D(S)$ for novelty and feasibility.

The results show that human researchers and GPT-4o on average rank the target research ideas in similar positions relative to the generated research ideas. From the average $D(S)$ we see 90% alignment between GPT-4o and humans for ranking the target papers for novelty, and 70% alignment for feasibility.

*Handling Potential Bias from GPT-4o as an Autorater.*

Google researchers show that LLMs can be used as reliable autoraters, and GPT-4o is overall the best off-the-shelf model in handling bias (Vu et al. (2024)). That's why we use GPT-4o as the autorater when evaluating research ideas. Furthermore, our relative quality metric does not prompt GPT-4o to give an absolute score for the quality of the ideas, because it may be biased. Rather, we provide a target idea to force GPT-4o to rank ideas based on a quality indicator specified by users, such as novelty and feasibility. This enables GPT-4o to provide a more objective evaluation than asking for an absolute score.

*Confidence Interval for Relative Quality Ranking.*

To ensure the robustness and consistency of our automatic evaluation, we calculated confidence intervals (CIs) for GPT-4o's relative quality rankings, which provide a clearer representation of the metric's reliability and variability. Using a dataset of m=100 target papers, we generated novel and feasible research ideas with our method and computed the average relative quality rankings (Average S ) across five iterations. This allowed us to obtain 95% confidence intervals for both novelty and feasibility, along with the standard deviation and variance.

|  | **Average $S$ CI** | **Standard Deviation** | **Variance** |
|---|---|---|---|
| **Novelty** | $0.983 \pm 0.003$ | 0.003 | $1.216 \times 10^{-5}$ |
| **Feasibility** | $0.484 \pm 0.026$ | 0.028 | $8.0464 \times 10^{-4}$ |

Table 5: Experiment assessing the consistency of GPT-4o's relative quality rankings. The table reports the 95% confidence intervals, standard deviations, and variances of the Average $S$ scores for novelty and feasibility, calculated five times.

The results, presented in Table 5, demonstrate that GPT-4o's rankings are highly consistent, with minimal variation in computed relative quality rankings, further supporting the validity of the metric.

*Comparison with Other Metrics.* In open-ended generation tasks, winrate is a metric commonly used to assess quality by determining the proportion of instances in which one model's output is preferred over another's in a binary comparison (Zheng et al., 2023). However, this approach reduces nuanced evaluations to binary outcomes, which can lead to significant information loss in capturing the diversity and subtle differences between outputs. Our relative quality ranking offers a more granular approach by allowing for a graded comparison across multiple dimensions of quality. Instead of a binary decision boundary, this metric ranks outputs on a continuum, capturing more nuanced differences in quality. This fine-grained assessment provides richer insights into the strengths and weaknesses of each model output, enhancing the accuracy of quality evaluations in open-ended generation tasks.

### A.3.2 EXPERIMENTS WITH OTHER MODELS

| Base Model | Average $S$ (Novelty) | Average $S$ (Feasibility) |
|---|---|---|
| Llamma 3.1 8B-Instruct | 0.953 | 0.451 |
| Llamma 3.1 70B-Instruct | 0.971 | 0.423 |
| Llamma 3.1 405B-Instruct | 0.988 | 0.363 |

Table 6: Main experiment results of our method with LLamma 3.1 family of models as the base models. We report the average of the relative quality ranking scores, denoted as Average S, for novelty and feasibility.

We conducted more experiments with LLamma 3.1 family of models as the base models of our method. We report the results in Table 6. We inform the readers that due to uncontrollable errors during API calls to generate research ideas with Llama 3.1 405B-Instruct, 167 data points for novelty and 64 for feasibility were discarded when evaluating the model. The results below show that open-sourced models can also benefit from our proposed method and achieve relatively high scores for generating research ideas.

### A.3.3 COST FOR DEPLOYMENT

| Base Model | Average Cost Per Idea |
|---|---|
| GPT-4o | $1.27 |
| GPT-4o Mini | $0.21 |
| GPT-3.5 Turbo | $0.88 |
| Llamma 3.1 405B-Instruct | $0.27 |
| Llamma 3.1 70B-Instruct | $0.04 |
| Llamma 3.1 8B-Instruct | $0.02 |

Table 7: Average cost of generating a research idea using our method with different backbone LLMs.

We calculated the average cost of generating a research idea for each model using our method and report the costs in 7. The cost were calculated with OpenAI [1] and DeepInfra [2] (API service for Llamma 3.1 models) cost per million tokens. We see that GPT-4o is the most expensive model to generate research ideas with, while Llamma 3.1 8B-Instruct is the cheapest.

### A.4 CASE STUDIES

In this section, we present two examples of how our method improves research ideation. One example focuses on improving the novelty of the research idea (Figure 14) and the other on feasibility (Figure 15). These examples show how zero-shot in-context adversarial learning with multi-LLM-agent interactions refines research ideas to improve novelty or feasibility.

Both examples begin with the proposer agent generating the same initial idea which hypothesizes that orexin levels may improve sleep quality and energy levels in adolescent athletes. When improving novelty, the reviewer agent suggests ways to make the idea more innovative. After seven iterations, the idea evolves to suggest that modulating orexin levels can enhance cognitive function and emotional resilience in adolescent athletes and proposes to use advanced statistical models to analyze the relationship between orexin, social jetlag, and cognitive/emotional outcomes to uncover insights that could redefine orexin's role.

---

[1]More details about the OpenAI API service can be found here: https://platform.openai.com/docs/overview

[2]More details about the DeepInfra's API service can be found here: https://deepinfra.com/

For feasibility, the process focuses on adding practical details without significantly changing the hypothesis. The reviewer agent suggests steps like using a manageable sample size and activity monitors for data collection. After seven iterations, the final idea proposes the hypothesis that simplified dietary guidelines and sleep hygiene education can improve sleep quality and energy levels in adolescent athletes. The study is designed to test this hypothesis with 15 participants, using validated questionnaires and activity monitors, and involving parents to ensure adherence to the study guidelines.

These case studies demonstrate the effectiveness of our method in refining research ideas through iterative multi-agent interactions. By focusing separately on novelty and feasibility, the process adapts initial ideas into more innovative or practical research ideas, highlighting the versatility of zero-shot in-context adversarial learning for enhancing the quality of research ideation.

## Example: Improving Novelty

**Iteration 1**

Hypothesis: The Impact of Orexin Modulation on Sleep Quality and Energy Availability in Adolescent Athletes** ... we hypothesize that modulation of orexin levels may improve sleep quality and energy availability in this population. Specifically, we propose that targeted interventions to stabilize orexin signaling can reduce sleep disturbances and mitigate the effects of LEA, thereby enhancing overall athletic performance and well-being. This hypothesis integrates the neuroendocrine insights regarding orexin's role with practical observations in adolescent athletes, offering a potential therapeutic avenue that addresses both metabolic and sleep-related challenges.

.... To improve the novelty of your research idea, consider the following: Explore less-studied aspects of orexin's role, such as its impact on cognitive functions or emotional well-being. ... Utilize cutting-edge techniques like optogenetics or CRISPR, or combine orexin modulation with other interventions. ... Propose a novel mechanism or interaction involving orexin that challenges existing paradigms. ... Identify unique population subsets or novel environmental factors to study. ...Redefine orexin's role in a broader context, such as its interaction with circadian rhythms.... Focus on long-term effects, gender differences, or other nuanced aspects of orexin modulation in adolescent athletes. ...

**Iteration 2**

... Hypothesis: Modulating orexin levels will improve cognitive function and emotional well-being in adolescent athletes, leading to enhanced athletic performance and better energy management.... Use advanced statistical methods to analyze the data, focusing on the interaction between orexin modulation, cognitive/emotional outcomes, and athletic performance. Expected Impact/Findings: .... Improved cognitive functions and emotional well-being in adolescent athletes. .... Enhanced athletic performance and better energy management .... New understanding of orexin's role in cognitive and emotional domains ....Insights into how orexin modulation differentially impacts male and female athletes...

The role of orexin in cognitive and emotional domains has been suggested in other contexts. To increase creativity, consider a more specific or novel angle. For example, hypothesize that orexin modulation could differentially affect specific cognitive domains (e.g., executive function vs. memory) or emotional states (e.g., anxiety vs. depression) in adolescent athletes. ... Innovate by incorporating emerging technologies or methodologies. For instance, use real-time monitoring of orexin levels through wearable biosensors or employ machine learning algorithms to personalize intervention strategies based on individual responses. ... To be more disruptive, propose a novel mechanism by which orexin influences cognitive and emotional outcomes. ...

{\"Is there a significant improvement?\": \"Yes\"}

⋮

**Iteration 7**

... Hypothesis: Modulating orexin levels will enhance cognitive function and emotional resilience in adolescent athletes by mitigating social jetlag and promoting synaptic plasticity, leading to improved performance and well-being.... Use advanced statistical methods and machine learning to analyze the data, focusing on the interaction between orexin modulation, social jetlag, synaptic plasticity, and cognitive/emotional outcomes.... Expected Impact/Findings: .... Enhanced cognitive function and emotional resilience in adolescent athletes.... Improved athletic performance and better energy management.... New understanding of orexin's role in mitigating social jetlag and promoting synaptic plasticity.... Development of personalized intervention strategies.... Insights into the long-term effects of orexin modulation.... Redefinition of orexin's role as a central integrator of social jetlag, synaptic plasticity, and mental health in adolescent athletes...

... Explore how orexin modulation might influence less-studied factors like social interactions and team dynamics or the impact of different types of physical training.
... Innovate by integrating novel technologies like non-invasive brain stimulation or using digital twin models to simulate individual responses.
... Propose a novel mechanism by which orexin influences outcomes, such as its impact on the gut-brain axis or neurogenesis.
... Focus on unique aspects of adolescent athletes, such as orexin's role in social hierarchies within teams or injury recovery.
... Propose a new conceptual framework where orexin serves as a central integrator of physical activity, sleep, and mental health.
... Focus on a more specific gap, like orexin's differential impact based on social support or its role in mitigating chronic stress and burnout in high-performing athletes.

{\"Is there a significant improvement?\": \"No\"}

Proposer Agent          Reviewer Agent          Area Chair Agent

Figure 14: An example of how the zero-shot in-context adversarial learning helps to improve the novelty of one generated research idea.

Figure 15: An example of how the zero-shot in-context adversarial learning helps to improve the feasibility of one generated research idea.

