# OpenReview forum: "Review and Rebuttal: Zero-shot In-context Adversarial Learning for Improving Research Ideation"
_ICLR.cc/2025/Conference — Submitted to ICLR 2025_

### Official Review · Reviewer_T15G · 2024-11-01

**Soundness:** 3
**Presentation:** 3
**Contribution:** 2
**Rating:** 5
**Confidence:** 4

**Summary:**

The paper introduces zero-shot in-context adversarial learning for Large Language Models (LLMs) to enhance research ideation by integrating adversarial learning techniques inspired by GANs. Through a system of multi-agent LLM interactions—including a proposer, reviewer, and area chair—the framework iteratively refines research ideas along novelty and feasibility dimensions. It uses a novel relative quality ranking metric to approximate human evaluations, offering scalable assessment of idea generation quality. The study shows substantial improvements, with a 21% boost in novelty and a 322% increase in feasibility for ideas generated by GPT-4, highlighting the potential of adversarial learning to enhance creativity and practical relevance in research ideation.

**Strengths:**

**Originality**
The paper adapts adversarial learning to zero-shot, in-context applications for LLM-driven research ideation. Using a multi-agent system modeled on academic peer review (proposer, reviewer, area chair), it effectively promotes iterative refinement in idea generation, filling a gap in the field with a conceptually novel approach.

**Quality**
The work is empirically solid, with extensive experimentation on a biomedical dataset. The model demonstrates clear performance gains in novelty and feasibility over strong baselines, with ablation studies and convergence analyses and potential for real-world application.

**Clarity**
The paper is well-organized, clearly explaining the framework, agent roles, and evaluation metrics.

**Significance**
The framework has potential for advancing automated scientific ideation, making research ideation more scalable and accessible. Its relative ranking metric could potentially generalize to other applications.


**Summary**
This paper is a valuable contribution in LLM-driven research and ideation.

**Weaknesses:**

The relative quality ranking metric would benefit from validation through a small-scale human study or comparisons with established evaluation metrics. This would improve confidence in the metric as a reliable proxy for human assessment, particularly for its scalability and alignment with human judgment.

Expanding experiments to other domains could demonstrate the framework’s adaptability beyond biomedical research, supporting its broader applicability claims.

Discussing practical deployment considerations for the multi-agent system, such as computational overhead.

**Questions:**

How reliable is the relative quality ranking metric as a proxy for human judgment in assessing novelty and feasibility? Has the metric been validated against human evaluations or other established metrics in a controlled way? Including these details or insights would strengthen confidence in the metric's robustness.

---

> ### Author Response · Authors · 2024-11-20
> **Authors' Response to Reviewer T15G**
>
> We sincerely thank Reviewer T15G for the constructive feedback and thoughtful comments. We hope you can first review our general response, where most of your points may already be addressed.
>
> ## Reply to weakness 1:
> We hope the Relative Quality Ranking section of the general response addresses these concerns.
>
> ## Reply to weakness 2:
> Thank you for your suggestion. While we agree that expanding experiments to additional domains could further illustrate the framework's adaptability, we argue that the diversity within our biomedical dataset already provides a robust demonstration of its broad applicability. The dataset spans a wide range of subfields, including cancer research, biology, neurology, and psychiatry. Each of these subfields demands specialized expertise, follows distinct disciplinary frameworks, and poses unique challenges for research ideation. By effectively navigating this diversity, our framework demonstrates its ability to adapt to varied contexts within a complex domain.
>
> We believe this diversity serves as a strong proxy for testing the framework’s generalizability. Future work could explore its application to entirely different domains, such as engineering or social sciences, to further substantiate its broader adaptability.
>
> ## Reply to weakness 3:
> We hope the Cost for Deployment section of our general response addresses this concern. Additionally, the appendix provides the implementation details necessary for someone to deploy our system.
>
> ## Reply to question 1
> We hope the Relative Quality Ranking section in the general response addresses these questions. In that section, we present a human study demonstrating the alignment of our metric with human rankings of research ideas, explain our rationale for selecting GPT-4o as the autorater to mitigate potential biases in evaluation, and provide the confidence interval for evaluating research ideas using our Relative Quality Ranking metric.

---

> > ### Author Response · Authors · 2024-12-02
> > **Final Opportunity for Feedback on Rebuttal**
> >
> > Dear Reviewer T15G,
> >
> > Thank you for taking the time to engage with us despite your busy schedule. As today marks the final day for submitting rebuttals, we hope you’ve had a chance to review our most recent response. We have made every effort to address your concerns thoroughly and thoughtfully.
> >
> > If possible, we kindly invite you to share any final thoughts or suggestions. Additionally, if you find that our revisions and clarifications have satisfactorily addressed your concerns, we would greatly appreciate it if you could consider updating your rating score accordingly.
> >
> > Your final input is invaluable in ensuring a fair and comprehensive review process. Thank you once again for your time and expertise throughout this discussion.
> >
> > Best regards,
> > Authors

---

### Official Review · Reviewer_eQ6T · 2024-11-03

**Soundness:** 3
**Presentation:** 3
**Contribution:** 3
**Rating:** 6
**Confidence:** 4

**Summary:**

This paper examines an approach that they call zero-shot in-context adversarial learning to enhance research ideas generated by LLMs. To evaluate this method, the authors introduce a relative quality ranking metric that assesses the quality of LLM-generated ideas against a benchmark of human-generated ideas. The metric focuses on two key quality indicators: novelty and feasibility.  Their work involves creating a dataset of 500 high-quality biomedical research papers, with the research ideas from these papers serving as the "gold standard" for both novelty and feasibility.  Target papers provide background information for the LLMs, simulating a human researcher gathering context before ideation. The system, using different LLMs (GPT-4o, GPT-4o Mini, GPT-3.5 Turbo), generates research ideas based on this background information.  GPT-4o is then tasked with ranking these LLM-generated ideas alongside the human-generated idea based on either novelty or feasibility. This ranking is done blindly, without GPT-4o knowing which idea is human-generated. The relative quality ranking score is then calculated based on the rank of the human-generated idea. A higher score indicates that the LLM-generated ideas are generally ranked higher (meaning better) than the human-generated one for the given quality indicator. Humans set the stage by providing the benchmark ideas and contextual information, while GPT-4o uses this information to evaluate and rank the quality of LLM-generated ideas. This allows for a quantifiable assessment of how well the zero-shot in-context adversarial learning method enhances the novelty and feasibility of research ideas compared to human performance.

**Strengths:**

The paper presents a novel approach to enhancing research ideation using LLMs.

Originality:

The paper examines the seemingly novel setting of "zero-shot in-context adversarial learning" specifically for research idea generation. This framework draws inspiration from GANs but adapts it to the unique challenges of working with LLMs and open-ended tasks. This indeed represents a creative combination of existing ideas applied to a new domain. The paper proposes a new evaluation metric, the "relative quality ranking score," to assess LLM-generated research ideas against a benchmark of human-generated ideas. This addresses the challenge of evaluating open-ended text generation, moving beyond traditional metrics and offering a more nuanced assessment. The authors implement the adversarial learning framework through a unique multi-agent system, where each agent (Proposer, Reviewer, Area Chair) plays a specific role in the idea refinement process. This mimics the dynamics of scientific peer review and leverages the strengths of multiple LLMs working in concert.

Quality:
The experiments demonstrate the effectiveness of the proposed method. The results show significant improvements in both novelty and feasibility of the generated ideas compared to baselines and human-generated ideas.  This highlights the practical value of the approach.
The paper provides a comprehensive analysis of the method's performance, including convergence analysis and ablation studies. This evaluation strengthens the claims made and provides insights into the contribution of each component of the system. The authors build a dataset of 500 high-quality biomedical research papers and their references, providing a robust foundation for evaluating the research idea generation process.

Clarity:
The theoretical framework of zero-shot in-context adversarial learning is clearly articulated – but it is important for the reader to appreciate that the GAN framing is really more of a metaphor since as the authors point out, there is no backprop going on and the theta’s used are  not parameters of the model. The paper provides reasonable explanations and illustrative examples of the agent interactions, prompt templates, and the relative quality ranking metric. This level of detail enhances reproducibility and transparency. The inclusion of case studies helps to demonstrate the practical application of the method and how it leads to improvements in both novelty and feasibility of research ideas. These examples make the benefits of the approach more tangible and accessible to readers.

Significance:
The paper contributes significantly to the growing body of research exploring the potential of LLMs in scientific discovery. The proposed method offers a promising avenue for leveraging LLMs to assist researchers in generating and refining high-quality research ideas.  The paper's theoretical foundation and empirical findings can contribute to a better understanding of in-context learning in LLMs, particularly how adversarial dynamics can enhance the utilization of LLMs' parametric knowledge, despite being a bit “hand wavy” in the way in which the mathematics here is being used to describe the method. The methods and evaluation techniques introduced in the paper could be adapted and applied to other domains involving user interaction with LLMs, potentially leading to improvements in areas such as creative writing, problem-solving, and decision-making.

Overall, the paper presents a fairly well-executed and clearly communicated study that introduces a novel and effective approach to enhancing research ideation with LLMs.

**Weaknesses:**

Weaknesses and Areas for Improvement

1. Limited Scope of Evaluation
The evaluation focuses solely on the biomedical domain. While the approach is theoretically applicable to other research areas, the generalizability of the findings to other domains needs to be investigated. Experiments with datasets from diverse research areas would strengthen the claims of broader applicability. But this is a somewhat minor issue in my view as this paper is the first to introduce this idea.
The current evaluation assesses novelty and feasibility separately, without considering their interplay. A combined metric or analysis of the trade-offs between these qualities would provide a more holistic view of the generated ideas' overall quality. Comparisons to other relevant baselines are limited. For instance, comparing against methods that specifically target novelty or feasibility in idea generation, such as those referenced in the related work section, might provide a more comprehensive assessment of the method's performance. While the relative quality ranking metric offers a scalable alternative, including a smaller-scale human evaluation study would provide valuable insights into the alignment between GPT-4o's rankings and human judgments of novelty and feasibility. This would strengthen the validity of the proposed metric as a proxy for human evaluation.

2. Potential Bias in Evaluation
 Using GPT-4o for both idea generation and evaluation introduces potential bias. While the ranking process is blinded, there's a possibility that GPT-4o might implicitly favor ideas generated by its own model family. Exploring alternative evaluation methods or incorporating human evaluators could mitigate this concern. The performance of the system is likely sensitive to the specific prompts used for each agent. A more in-depth analysis of prompt engineering techniques and their impact on the quality of generated ideas would be beneficial.
* How robust is the method to variations in prompting?
* What are the best practices for crafting effective prompts?

3. Computational Cost and Efficiency
The multi-agent system, especially when using high-capacity LLMs like GPT-4o, likely requires substantial computational resources. A discussion on the computational cost and potential optimizations for efficiency would be valuable for practical implementation.

4. Theoretical Limitations
The GAN formulation is mathematical, but more metaphorical than actually a rigorous description of the procedure here: This is the biggest weakness in my view. This GAN framing starts out seeming conceptually coherent with the procedure that is going to be applied, but the way in which the thetas and theta stars are used to define the procedure of altering parametric memories just doesn’t seem coherent with what is actually being done here. The whole procedure is more of a dialogue and in the end it is about generated tokens and their properties, and not really about parametric memory updates. This part of the theoretical presentation is weak and it makes it hard to also perform any real theoretical analysis because it just doesn’t seem consistent with what is actually being done.

5. The experiments:
The paper compares the proposed method against two baselines: the initial idea baseline and the self-reflection baseline. While these baselines provide a starting point, including additional baselines that represent alternative approaches to LLM-based idea generation would strengthen the evaluation. For example, comparing against methods that use prompt engineering (e.g. DSPy) or fine-tuning techniques for research idea generation would provide a more comprehensive assessment of the proposed method's effectiveness.

6. Future Directions
User Interaction and Feedback: The system, in its current form, assumes a fixed set of quality indicators (novelty and feasibility). Exploring mechanisms for incorporating user preferences and feedback into the refinement process would enhance the system's usability and tailor it to specific research needs.
* How could the system be made more interactive and responsive to user input?

**Questions:**

Questions and Suggestions for the Authors

1)	Could the theoretical explanation of this whole procedure be improved and synchronized more with the reality of the method? I might be missing something, but I find this whole theta and theta star framing to be a bit of a distraction from what seems to really be going on. I open to being convinced that this way of thinking about it is coherent with the actual procedure here, but I think this could be reformulated to make the paper a much more significant contribution.

2)	An idea might be very novel but completely infeasible.
How does the system handle this tension?

3)	Regarding the Novelty of Relative Quality Ranking Metric: While the paper introduces the relative quality ranking metric as a novel contribution, a discussion on its relationship to existing evaluation metrics for open-ended text generation would be beneficial. Are there similar metrics in the literature? How does the proposed metric offer advantages or address limitations of these existing metrics?

4)	The paper acknowledges the potential of the proposed method for other tasks involving LLM interaction. However, providing concrete examples of such tasks and discussing how the framework could be adapted would strengthen the claims of broader impact and significance. What specific adaptations or modifications would be needed to apply the method to other tasks?

5)	Regarding the Area Chair:
Why do you think the removal of the area chair agent has a more pronounced impact on feasibility compared to novelty?
Do you have any insight on the specific cues or signals the Area Chair agent might be looking for to determine whether significant improvements have been made ?
Have you considered examining how the agent's judgment aligns with human assessments of improvement?

---

> ### Author Response · Authors · 2024-11-20
> **Authors' Response to Reviewer eQ6T (part1/2)**
>
> We sincerely thank Reviewer eQ6T for the constructive feedback and thoughtful comments. Below, we address your points in detail. We hope you can review our general response, where several of your points may have already been addressed.
> ## Reply to weakness 1
> Our biomedical dataset encompasses studies from diverse subfields, ranging from cancer and biology to neurology and psychiatry. Since each subfield requires distinct expertise and follows unique disciplinary approaches, we argue that the dataset’s diversity is sufficient to showcase the effectiveness of our proposed method.
>
> To make sure our system can help optimize the quality of ideas from different dimensions, we decompose the trait of a "good idea" into novelty and feasibility and test them seperately. But in practice, the users may have different requirements for different quality indicators, so the optimization process is highly customizable by tweaking the qaulity indicators.
>
> We hope the Relative Quality Ranking section in general response can address other concerns in this point.
>
>
> ## Reply to weakness 2
> In this work, we mainly focus on implementing the logic of in-context adversarial learning so we just make sure that all the related prompts reflected the system's logic. In our opinion, if naively crafted prompts can work, it will emphasize on the effectiveness of our proposed in-context adversarial learning. But we agree that the best practices for crafting prompts to best reflect the system's logic is worthwhile for future study.
>
> We hope the Relative Quality Ranking section in the general response can address other concerns in this point.
>
> ## Reply to weakness 3
> We hope the Cost for Depolyment in the general response can address this point.
>
> ## Reply to weakness 4
> Current theories on in-context learning largely draw upon metaphors from traditional machine learning theories. While these metaphors might not be strictly proven, they are supported by empirical evidence. For instance, in [TextGrad], the authors conducted extensive experiments to demonstrate that automatic differentiation can be effectively emulated through textual feedback (textual gradient) in LLMs, leading to improvements across various downstream tasks.
>
> In our setting, $\theta$ refers to the model's parametric knowledge rather than the actual model parameters. Since the learning process occurs within the context, the model's parameters remain fixed. Within our objective function, the reviewer agent provides textual feedback, which serves as "textual gradients" to guide the proposer agent in optimizing its exploration of the parametric knowledge base   {$\{\theta\}$}   to refine the generated idea $\hat{y}$. In other words, it's not updating the memories, it's optimizing the search in the model's parametric knowledge base to get better parametric knowledge. Upon convergence of this process, we identify $\theta^*$ within {$\{\theta\}$}.
>
> [TextGrad]Yuksekgonul, M., Bianchi, F., Boen, J., Liu, S., Huang, Z., Guestrin, C., & Zou, J. (2024). TextGrad: Automatic" Differentiation" via Text. arXiv preprint arXiv:2406.07496.
>
> ## Reply to weakness 5
> We appreciate the suggestion and agree that future work could explore comparisons with other methods to extend this research. However, for the scope of this paper, we believe that our chosen baselines effectively illustrate the strengths and contributions of the proposed method. Our framework is designed to maximize the potential of zero-shot in-context learning without requiring prompt engineering or fine-tuning. Comparing against methods that rely on carefully crafted prompts or model modifications would shift the focus from our core contribution, which lies in optimizing the utilization of LLM's parametric knowledge without external dependencies.
>
> ## Reply to weakness 6
> Thank you for the excellent suggestion. Our initial objective was to fully automate the research ideation process, designing the system to function in an end-to-end manner. However, this does not preclude customization. Since the interaction between the three agents occurs entirely through text, users can seamlessly take on the roles of the reviewer or area chair in this process. Additionally, users can input an initial idea and leverage the system to help refine and enhance it.

---

> > ### Author Response · Authors · 2024-11-20
> > **Authors' Response to Reviewer eQ6T (part2/2)**
> >
> > ## Reply to question 1
> > We hope our reply to weakness 4 can address this point. But feel free to let us know if you have further questions.
> >
> > ## Reply to question 2
> > The system can jointly optimize novelty and feasibility together if the users specify both of them in the placeholder {$\{quality indicator\}$}. We agree that the idea might be very novel but completely infeasible, but in practice, as different users may have different trade-off preferences, it's the users' decision for how to trade-off quality indicators like novelty and feasibility.
> >
> > ## Reply to question 3
> > We hope the Relative Quality Ranking section, especially the Comparison with Other Metrics subsection in general response can address this point.
> >
> > ## Reply to question 4
> > The proposed framework can be adapted to a variety of tasks. For instance, creative story writing, code generation, and product design ideation. Adapting the framework to these tasks would primarily involve redefining the agents' objectives and quality indicators (e.g., novelty, coherence, effectiveness, or efficiency) in each agent's prompt templates to suit the task's requirements. The core interaction mechanism remains applicable across domains.
> >
> > ## Reply to question 5
> > As we can see from our ablation study, the initial idea already has a very high relative quality ranking score for novelty but has a relatively low score for feasibility, this means the feasibility has more space to be improved. As the area chair agent plays a crucial role in evaluating whether a generated idea $\hat{y}$ has made consistent improvement compared to previously generated ideas, it has a more pronounced impact on feasibility. The cues and signals used by the area chair agent to evaluate improvements are the set of quality indicators defined in the prompt template. In the Relative Quality Ranking section, especially the Alignment with Human Judgment subsection in general response, we can see that relative quality ranking from GPT-4o aligns well with human judgement, and our results show that out proposed method can improve research ideas with respect to relative quality ranking. This indicates that area chair agent's judgment aligns with human assessments of improvement.

---

> > ### Comment · Reviewer_eQ6T · 2024-11-26
> > **Maintain my positive score**
> >
> > Thanks for your author response.
> > I am fine with what you have provided and I hold firm to my score of 6 and look forward to discussing this paper with other reviewers. I think this is interesting work that might stimulate more interesting discussions at ICLR.

---

> > > ### Author Response · Authors · 2024-11-27
> > > **Thank You for Your Positive Feedback**
> > >
> > > Dear Reviewer eQ6T,
> > >
> > > Thank you very much for your encouraging comments on our work and for maintaining your positive score. We deeply appreciate your thoughtful evaluation, constructive feedback, and your recognition of the potential for our work to spark further interesting discussions at ICLR.
> > >
> > > We are delighted that you found our responses satisfactory and our work engaging. If there are any additional thoughts or suggestions you would like to share, we would be happy to incorporate them to further strengthen the manuscript.
> > >
> > > Thank you once again for your time and support. We look forward to any discussions you may have with other reviewers regarding our work.
> > >
> > > Best regards,
> > > Authors

---

### Official Review · Reviewer_sPFq · 2024-11-05

**Soundness:** 2
**Presentation:** 2
**Contribution:** 2
**Rating:** 3
**Confidence:** 4

**Summary:**

This paper proposes an adversarial setting between LLM agents to do scientific ideation via iterative prompting. The setting includes a proposer, a review, and an area chair (AC) agent. The paper claims that the proposer and the AC serve as the role of generator and discriminator respectively just as the two roles in the traditional GAN setting. Each iteration, the proposer comes up with new ideas that are criticized by the reviewer agent, modified by the proposer again, and finally evaluated by the AC agent. Through multiple iterations until convergence or a hard limit, the system would be expected to produce a novel and feasible idea for a given user query. The paper also designs a new ranking-based metric with GPT-4o as a judge to evaluate idea quality. The paper conducts experiments with biomedical papers from semantic scholars and present good performance of the proposed method.

**Strengths:**

The paper adopts an interesting concept of adversarial learning from previous literature about GAN. The illustration and figures of the setting and pipeline is clear and intuitive. The structure of the paper is complete and the writing is coherent. Prompts for different agents are well-documented in the appendix. The paper overall is easy to read.

**Weaknesses:**

1. The setting though sounds interesting lacks mathematical foundation. While the original methods from GAN is well-established from theory to experiments, this paper adopts the concept of the minimax objective without implementing it with mathematical justifications. The discriminator relies on the assumption that the proposed ideas lies within neighborhood $B_\epsilon(y)$, which may not be realistic in practice. The performance of the reviewer is unclear. We do not know how close it is approximating a gradient update to guide the proposer to update generations. Multiple quality indicator traits are designed in the prompt but not evaluated specifically.

2. The evaluation is weak. The paper poses to evaluate LLM generations with LLM, which may carry neglected biases [1]. The ranking-based metric only reflects the relative quality of generated ideas, and lacks comparability across different batch or with other quality metrics. Both the proposed method and the validity of the metric needs user study for verification.

3. Experiments are not solid enough. All of the numbers reported lack confidence interval. There is no guarantee that the reported results is reproducible. All the models the paper evaluates the proposed methods on are from the OpenAI GPT family, which is not convincing enough. More models including open models should be in the experiments.

[1] Panickssery, Arjun, Samuel R. Bowman, and Shi Feng. "Llm evaluators recognize and favor their own generations." arXiv preprint arXiv:2404.13076 (2024).

**Questions:**

1. Line 206 where you mention $\hat{y}_{i-1} < \hat{y}_i$, what value are you comparing?
2. For each target paper, how do you select the $k$ reference papers as background information? Do you only consider papers cited by the target paper?
3. Do you have any observations of the reliability of your GPT-4o evaluation? Do they give the same ranking each time?

---

> ### Author Response · Authors · 2024-11-20
> **Authors' Response to Reviewer sPFq**
>
> We sincerely thank Reviewer sPFq for the constructive feedback and thoughtful comments. Below, we address your points in detail. For additional context or clarification, we hope you can first review our general response, where several of your points may have already been addressed.
>
> ## Reply to weakness 1
> Current theories on in-context learning largely draw upon metaphors from traditional machine learning theories. While these metaphors might not be strictly proven, they are supported by empirical evidence. For instance, in [TextGrad], the authors conducted extensive experiments to demonstrate that automatic differentiation can be effectively emulated through textual feedback (textual gradient) in LLMs, leading to improvements across various downstream tasks.
>
> In our ablation study, we demonstrated that removing the reviewer agent leads to a noticeable drop in the relative quality ranking and results in slower convergence of the entire system. This underscores the effectiveness of the reviewer agent in providing the "textual gradient" necessary for optimizing the generated outputs.
>
> For the evaluation of quality indicators, we adopted a divide-and-conquer approach. Specifically, we independently assessed novelty and feasibility to mitigate any intertwined effects, thereby showcasing the system's capability to optimize the quality of generated ideas across distinct dimensions.
>
> We noticed that our previous discussion about the area chair agent in Section 3.1.3 may cause confusion. Now we updated Section 3.1.3 and highlighted in red. Could you please review the latest Section 3.1.3 and let us know if the latest version clarifies the area chair's role better?
>
> [TextGrad]Yuksekgonul, M., Bianchi, F., Boen, J., Liu, S., Huang, Z., Guestrin, C., & Zou, J. (2024). TextGrad: Automatic" Differentiation" via Text. arXiv preprint arXiv:2406.07496.
>
> ## Reply to weakness 2
> We hope the Relative Quality Ranking section in general response can address this point. We provided a detailed user study, verifying that GPT-4o's judgment aligns well with human evaluators' judgment. We also provided more discussions on relative quality ranking and compared our metric with winrate metric. We updated the manuscript based on your suggestions, and the adjustments are highlighted in red.
>
> ## Reply to weakness 3
> We hope the Relative Quality Ranking section and More Experiments with Other Models sections in the general response can address this point. We added experiments with LLama model families and added the confidence interval of relative quality ranking. The results show that our proposed method can help open-source models achieve close to GPT-4o's result. We hope. The effectiveness of our proposed method is now more convincing.
>
> ## Reply to question 1
>
> As we mentioned in our paper around line 260, $\hat{y}_{i-1} < \hat{y}_i$
> indicates that significant improvements between the new idea $\hat{y}_i$
>
> and the previous idea $\hat{y}_{i-1}$ are identified by the area chair agent. As $\hat{y}$ represents an idea, it's not a concrete value, so the "$<$" stands for the quality of the idea on the right is better than that of the idea on the left. We've updated Section 3.1.3 to make it clearer.
>
> ## Reply to question 2
> Yes, we only consider papers cited by the target paper as background information to ensure a fair comparison. If we include related papers that are not cited by the target paper, it may help the proposer to generate better ideas but it's not fair for getting the relative quality ranking, as the idea from the target paper is generated based on the papers it cites.
> ## Reply to question 3
> We hope the Relative Quality Ranking section in general response can address this point. The confidence interval shows that using GPT-4o to perform relative quality ranking is very robust.

---

> > ### Author Response · Authors · 2024-12-02
> > **Final Opportunity for Feedback on Rebuttal**
> >
> > Dear Reviewer sPFq,
> >
> > Thank you for taking the time to engage with us despite your busy schedule. As today marks the final day for submitting rebuttals, we hope you’ve had a chance to review our most recent response. We have made every effort to address your concerns thoroughly and thoughtfully.
> >
> > If possible, we kindly invite you to share any final thoughts or suggestions. Additionally, if you find that our revisions and clarifications have satisfactorily addressed your concerns, we would greatly appreciate it if you could consider updating your rating score accordingly.
> >
> > Your final input is invaluable in ensuring a fair and comprehensive review process. Thank you once again for your time and expertise throughout this discussion.
> >
> > Best regards,
> > Authors

---

### Author Response · Authors · 2024-11-20
**General Response to All Reviewers (part1/2)**

We sincerely thank all the reviewers for their valuable suggestions and constructive feedback. Below, we aim to highlight several key points to provide greater clarity and a more comprehensive understanding of our work. We also update our paper to reflect these points.

# Relative Quality Ranking
## Alignment with Human Judgment
In [SCIMUSE], the authors collaborated with over 100 research group leaders across diverse domains to rank more than 4,400 research ideas generated by their SCIMUSE system. Their findings revealed that LLM-based ranking, specifically using GPT-4o, aligns closely with human expert evaluations, achieving a top-1 precision of 51% and a top-5 precision of 46.7%. These results highlight the feasibility of using LLM-driven ranking as a scalable proxy for human evaluation, particularly when assessing large volumes of research ideas across various fields.

[SCIMUSE]Gu, X., & Krenn, M. (2024). Generation and human-expert evaluation of interesting research ideas using knowledge graphs and large language models. arXiv preprint arXiv:2405.17044.

To evaluate the alignment between GPT-4o and humans in assessing research ideas, we conducted a human study. We selected 10 sets of research ideas focused on novelty and 10 sets focused on feasibility, generated using our proposed adversarial in-context learning. Each set included three generated ideas and their respective target paper idea.

We recruited 10 researchers to rank the ideas in each set based on either novelty or feasibility, depending on the focus. The researchers were unaware of which ideas were generated and which originated from the target paper. We then compared the difference between relative quality ranking given by human researchers and GPT-4o $D(S)$:

$$D(S) =|S_{\text{Human}} - S_{\text{GPT-4o}}|$$
where $S_{\text{Human}}$ is the relative quality ranking from human researchers calculated using Formula (3) defined in our paper and similarly, $S_{\text{GPT-4o}}$ is the relative quality ranking from GPT-4o.

The following table shows the average $D(S)$ for novelty and feasibility:


|             | Average $D(S)$ $\downarrow$ |
| ----------- | --- |
| Novelty     | 0.1 |
| Feasibility | 0.3 |




This shows that human researchers and GPT-4o on average rank the target research ideas in similar positions relative to the generated research ideas. From the average $D(S)$ we see 90% alignment between GPT-4o and humans for ranking the target paper for novelty, and 70% alignment for feasibility.




## Handling Potential Bias from GPT-4o as an Autorater
The study from Google we cited shows that LLMs can be used as reliable autoraters, and GPT-4o is overall the best off-the-shelf model in handling bias [Foundational Autoraters]. That's why we use GPT-4o as the autorater in this work. Furthermore, we didn't ask GPT-4o to give an absolute score for the quality of the ideas, because it may be biased. Rather, we provide a target idea to force GPT-4o to rank all the ideas based on a quality indicator specified by the users like novelty and feasibility, which are more objective.

[Foundational Autoraters]Vu, T., Krishna, K., Alzubi, S., Tar, C., Faruqui, M., & Sung, Y. H. (2024). Foundational autoraters: Taming large language models for better automatic evaluation. arXiv preprint arXiv:2407.10817.
## Confidence Interval for Relative Quality Ranking
To ensure robustness, we incorporated confidence intervals into our relative quality ranking metric. This addition provides a clearer representation of the metric's reliability and variability, further supporting its validity.

To evaluate the consistency of GPT-4o’s relative quality rankings, we generated novel and feasible research ideas using our method with a dataset of $m=100$ target papers. We computed the average relative quality rankings (Average $S$) five times to obtain 95% confidence intervals (CIs) for novelty and feasibility, along with the standard deviation and variance:

|             | Average $S$ CI    | Standard Deviation | Variance                |
| ----------- | ----------------- | ------------------ | ----------------------- |
| Novelty     | $0.983 \pm 0.003$ | $0.003$            | $1.216 \times 10^{-5}$  |
| Feasibility | $0.484 \pm 0.026$ | $0.028$            | $8.0464 \times 10^{-4}$ |

The results demonstrate that GPT-4o’s rankings are highly consistent, with minimal variation in computed relative quality rankings.

---

> ### Author Response · Authors · 2024-11-20
> **General Response to All Reviewers (part2/2)**
>
> ## Comparison with Other metrics
> In open-ended generation tasks, winrate is a metric commonly used to assess quality by determining the proportion of instances in which one model's output is preferred over another's in a binary comparison [MT-Bench]. However, this approach reduces nuanced evaluations to binary outcomes, which can lead to significant information loss in capturing the diversity and subtle differences between outputs. Our relative quality ranking offers a more granular approach by allowing for a graded comparison across multiple dimensions of quality. Instead of a binary decision boundary, this metric ranks outputs on a continuum, capturing more nuanced differences in quality. This fine-grained assessment provides richer insights into the strengths and weaknesses of each model output, enhancing the accuracy of quality evaluations in open-ended generation tasks.
>
> [MT-Bench]Zheng, L., Chiang, W. L., Sheng, Y., Zhuang, S., Wu, Z., Zhuang, Y., ... & Stoica, I. (2023). Judging llm-as-a-judge with mt-bench and chatbot arena. Advances in Neural Information Processing Systems, 36, 46595-46623.
>
> # More Experiments with Other Models
> We conducted more experiments with the LLama 3.1 family of models. The results below show that open-sourced models can also benefit from our proposed method and achieve relatively high scores for generating research ideas.
>
> | Model                    | Average S (Novelty) | Average S (Feasibility) |
> | ------------------------ | ------------------- | ----------------------- |
> | Llamma 3.1 8B-Instruct   | 0.953               | 0.451
> | Llamma 3.1 70B-Instruct  | 0.971               | 0.423                   |
> | Llamma 3.1 405B-Instruct | 0.988               | 0.363                   |
>
>
>
>
> # Cost for Deployment
>
>
> For your reference, we also calculated the average cost to generate an idea for each model using our proposed method.
> | Backbone LLM             | Average Cost Per Idea |
> | ------------------------ | ---------------------------------------- |
> | GPT-4o                   | $1.27                                    |
> | GPT-4o Mini              | $0.21                                    |
> | GPT-3.5 Turbo            | $0.88                                    |
> | Llamma 3.1 405B-Instruct | $0.27                                        |
> | Llamma 3.1 70B-Instruct  | $0.04                                    |
> | Llamma 3.1 8B-Instruct   | $0.02                                    |

---

> > ### Author Response · Authors · 2024-11-24
> > **Updated Manuscript with Additional Experiments and Discussions**
> >
> > Dear Reviewers,
> >
> > We want to inform you that we have updated our manuscript based on the discussions and feedback provided. Specifically, we have added additional experimental results and in-depth discussions in the appendix to address the points raised.
> >
> > We greatly appreciate your valuable insights and suggestions, which have significantly contributed to improving our work. We welcome any further questions or recommendations you may have to help us enhance the manuscript even more.
> >
> > Thank you for your time and consideration.
> >
> > Best regards,
> > Authors

---

> > > ### Author Response · Authors · 2024-11-26
> > > **Highlighted Edits and Open Discussion Before Manuscript Update Deadline**
> > >
> > > Dear Reviewers,
> > >
> > > As the deadline for authors to update the manuscript approaches, we would like to inform you that all newly edited sections in our manuscript have been highlighted in red to make it easier for you to review the changes.
> > >
> > > We highly encourage you to engage in discussions with us regarding any remaining concerns or questions. We are eager to address your feedback and reflect any additional suggestions in the manuscript before the deadline. Your insights and guidance are invaluable in ensuring the quality and clarity of our work.
> > >
> > > Thank you for your time and effort in reviewing our submission. We greatly appreciate your support and look forward to hearing from you.
> > >
> > > Best regards,
> > > Authors

---

### Meta-Review · Area_Chair_Z2Cb · 2024-12-20

**Metareview:**

The authors propose a GAN-inspired approach to the generation of scientific ideas - instead of taking parametric updates, the approach put the updates into the context of the model, using in-context-learning as the update mechanism. On the empirical side, the authors use LLM-evals to evaluate the quality of the ideas and show that the proposed algorithm improves on this metric.

The reviewers are fairly unanimous on two fairly important points - as a theory work, the 'prompt to do a GAN' approach is lacking, as there's no real formal model or optimization problem that's being closely approximated. One could argue all of math in ML is just an intuition-building mechanism, but those papers need to show their worth through empirical results. The other issue is the deep reliance on automatic evals for a very tricky and subjective eval setting such as ideation - multiple reviewers point out the problems with this approach, and weakens the empirical side of this work.

**Additional Comments On Reviewer Discussion:**

Reviewers and authors engaged in some clarifications regarding the mathematical formalism and empirical validity.

---

### Decision · Program_Chairs · 2025-01-22

Reject